# Transcriptomic Evidence for Cell-Autonomous Sex Differentiation of the Gynandromorphic Fat Body in the Silkworm, *Bombyx mori*

**DOI:** 10.3390/jdb12040031

**Published:** 2024-11-20

**Authors:** Fumiko Yamamoto, Takeshi Yokoyama, Yan Su, Masataka G. Suzuki

**Affiliations:** 1Anicom Pafe, Inc., Sumitomo Fudosan, 8-17-1, Shinjyuku, Shinjyuku-ku 160-0023, Tokyo, Japan; fumimoto365@gmail.com; 2Department of Biological Production, Faculty of Agriculture, Tokyo University of Agriculture and Technology, 3-8-1, Harumi-cho, Fuchu 183-8538, Tokyo, Japan; ty.kaiko@cc.tuat.ac.jp; 3Department of Physiology, Graduate School of Medicine, Juntendo University, 2-1-1 Hongo, Bunkyo-ku 113-8421, Tokyo, Japan; yansu1995-2020@outlook.com; 4Department of Integrated Biosciences, Graduate School of Frontier Sciences, The University of Tokyo, 5-1-5, Kashiwanoha, Kashiwa 277-8562, Chiba, Japan

**Keywords:** sex determination, cell-autonomous regulation, secondary sexual traits, *Feminizer*, fat body

## Abstract

The classic model of sex determination in insects suggests that they do not have sex hormones and that sex is determined in a cell-autonomous manner. On the other hand, there is accumulating evidence that the development of secondary sexual traits is controlled in a non-cell-autonomous manner through external factors. To evaluate the degrees of the cell-autonomous and non-cell-autonomous regulation of secondary sexual trait development, we analyzed the dynamics of the sexually dimorphic transcriptome in gynandromorphic individuals of the *mo* mutant strain in the silkworm *Bombyx mori*. The silkworm possesses a female heterogametic sex-determination system (ZZ = male/ZW = female), where the master regulatory gene for femaleness, *Feminizer* (*Fem*), is located in the W chromosome. As a secondary sexual trait, we focused on the fat body, which shows remarkable differences between the sexes during the last instar larval stage. A comparison of the transcriptomes between the fat bodies of male and female larvae identified 232 sex-differentially expressed genes (S-DEGs). The proportions of ZZ and ZW cells constituting the fat body of the gynandromorphic larvae were calculated according to the expression level of the *Fem*. Based on the obtained values, the expression level of each S-DEG was estimated, assuming that the levels of S-DEG expression were determined according to the proportion of ZZ and ZW cells. The estimated expression levels of 207 out of 232 S-DEGs were strongly correlated with the corresponding S-DEG expression level of the gynandromorphic fat body, determined by RNA-seq. These results strongly suggest that most of the sexually dimorphic transcriptome in the fat body is regulated in a cell-autonomous manner.

## 1. Introduction

In many vertebrate species, sex hormones secreted from the gonads play a pivotal role in sexual differentiation. The sexual fate of the gonads in mammals is determined by the action of the sex-determining region Y (SRY) gene located in the Y chromosome, which triggers the undifferentiated gonads to develop as a testis [1,2,3]. Once they have developed, the gonads secrete sex hormones, such as estrogen and testosterone, leading to the sexual development of the body. Many species of fish have high plasticity in sexual differentiation, and the administration of androgens and estrogens during early juvenile stages can cause sex reversal, even after the genetic sex has been determined [4,5]. Thus, sexual differentiation in vertebrates is strongly dependent on sex hormones and is generally considered to be regulated in a non-cell-autonomous manner.

The first report of the possible presence of sex hormones in insects was made by Naisse regarding the firefly *Lampyris noctiluca* [6]. The transplantation of the testis into female larvae was shown to induce maleness, and it was concluded that the apical tissue in the testis may function as a male gland. However, these findings were controversial as similar transplantation experiments by another group failed to reproduce these results [7]. Since then, there have been no reports clearly indicating the presence of sex hormones in insects.

On the other hand, the mechanisms of the cell-autonomous regulation of sexual development are well understood in insects. The *doublesex* gene (*dsx*) sits at the bottom of the sex determination cascade and acts as a master regulatory gene for sex differentiation in all insect species investigated to date [8]. The pre-mRNA of the *dsx* gene undergoes sexually dimorphic alternative splicing, producing male- and female-specific isoforms (*dsxM* and *dsxF*) that promote male and female development, respectively [9,10,11,12,13,14]. Functional analyses of *dsx* in various species showed that sexual dimorphism can be altered simply by switching the expression of male and female isoforms of *dsx* [15,16,17,18,19,20,21,22]. Moreover, somatic mutations in *dsx* induced by genome editing techniques, such as CRISPR/Cas9 and TALEN, caused male–female mosaicism in various sexually dimorphic traits [13,23,24]. These observations strongly suggest that the development and differentiation of sexually dimorphic traits in insects are regulated in a cell-autonomous manner by the sex-specific isoforms of *dsx*, the expression of which is under direct genetic control in individual cells.

The spontaneous appearance of gynandromorphic individuals is the strongest evidence in support of the cell-autonomous regulation of sexual differentiation in insects [25,26,27,28,29,30]. Gynandromorphic individuals consist of cells with different genetic sexes, and exhibit both male and female sexual phenotypes in a mosaic pattern, with the most dramatic case being male and female external phenotypes fused symmetrically at the midline of the body [26,27,28]. Sexual characteristics in the internal organs, such as the gonads and central nervous system, and physiological sexual characteristics, such as body size and vitellogenin biosynthesis, are also lateralized in symmetrical male and female mosaic animals. Theoretically, gynandromorphism cannot appear when humoral factors, such as sex hormones, control sexual differentiation. Based on these findings, it has long been believed that insects do not have sex hormones, and that their sex is determined cell-autonomously.

However, studies focusing on secondary sexual traits have yielded evidence that sexual differentiation in insects is regulated by hormonal factors. In *Drosophila melanogaster*, ecdysone (E) signaling plays an important role in the modification of male and female courtship behaviors, in addition to cell-autonomous regulation through *dsx* and *fruitless* (*fru*) [31,32]. Similarly, feeding males with an active form of E, 20-hydroxyecdysone (20E), exhibited female behavior in the case of the blowfly *Sarcophaga bullata* [33]. The takeout protein, which shows sequence similarity to the juvenile hormone binding protein, is specifically synthesized in male fat bodies and is secreted into the hemolymph to promote male-specific behavior by acting in the brain [34,35]. Yolk protein synthesis in the *Drosophila* fat body is specifically activated in females by *dsxF* and is repressed in males by *dsxM* [36]. Interestingly, previous work showed that the injection of 20E stimulated the process of yolk protein synthesis in males [37]. In the housefly *Musca domestica*, ecdysteroid titer in the hemolymph differs between males and females during the oogenic phase, and a higher ecdysteroid level in females is correlated with higher levels of yolk proteins in the hemolymph [38,39]. The E/20E ratio in several lepidopteran insects is higher in males than females from the beginning of the pupal stage due to the predominance of E in males [40]. In the silkworm *Bombyx mori*, an insulin-like growth factor-like peptide secreted from the fat body, brain, and gonads shows higher titer in males than females during pupal stages [41,42,43]. However, it remains unclear as to whether this peptide is important for sexually dimorphic trait development in silkworms. Based on these findings, Prakash and Monteiro suggested that the formation of sexually dimorphic traits in insects is equally regulated by both cell-autonomous and non-cell-autonomous mechanisms [24].

Femaleness in silkworms is determined by the presence of the W chromosome, as *Feminizer* (*Fem*), the master regulatory gene for femaleness, is carried in this chromosome [44,45]. *Fem* transcripts cause the degradation of transcripts from the *Masculinizer* gene (Masc), which plays a pivotal role in maleness [46,47]. In the absence of Masc expression, *Bombyx doublesex* (*Bmdsx*) pre-mRNA undergoes female-specific splicing to produce the female-type isoform of *Bmdsx*, *BmdsxF*, which enhances female development [11,45,47,48]. In males, *Masc* is stably expressed due to the lack of the W chromosome, resulting in the expression of the male-specific isoform of the *Bombyx* homologue of the insulin-like growth factor II mRNA-binding protein gene (*ImpM*) [45,48]. *ImpM* leads to the production of the male-type isoform of *Bmdsx*, *BmdsxM*, which promotes male development [12,49].

In this study, we attempted to quantify the integrity of secondary sexual trait development in gynandromorphic individuals to clarify the degrees of contribution of cell-autonomous and non-cell-autonomous regulation to sexually dimorphic trait development in insects. For this purpose, we used a silkworm strain that frequently yields gynandromorphic animals. As an insect secondary sexual trait, we focused on the fat body because remarkable sexual dimorphisms in the fat body, such as yolk protein levels and vitellogenin biosynthesis, are regulated by both cell-autonomous and non-cell-autonomous mechanisms, as outlined above. The fat body is a tissue that combines the functions of both the adipose tissue and the liver in mammals. The fat body synthesizes a variety of physiologically active substances and serves as a center for metabolic control and immune response [50,51]. The fat body is composed of flattened tissue anchored by partial attachment to the inner walls of the epidermis and is distributed throughout the body in a suspended state. This floating structure of the fat body facilitates interaction with the hemolymph. A comparison of the transcriptomes between male and female fifth instar larval fat bodies was conducted to perceive the whole picture of sexual dimorphism in this tissue at the molecular level, which consequently identified 232 sex-differentially expressed genes (S-DEGs). The sexually dimorphic status of S-DEG expression was investigated in the fat body under various conditions, such as gynandromorphic status and culture in vitro using hemolymph from male and female larvae. The data obtained in this study suggest that the sexually dimorphic expression of more than 90% of all S-DEGs is regulated cell-autonomously under the sex-determining cascade according to the primary genetic signal in individual cells. To our knowledge, this study provides the first transcriptomic evidence for cell-autonomous sex differentiation of secondary sexual traits in an insect species.

## 2. Materials and Methods

### 2.1. Silkworm Strains

The silkworm strain m042 carrying the mo (hereditary mosaic) mutation, females of which produce mosaic and gynandromorphic offspring at high rates (Appendix A) [52,53], was provided by the Silkworm Stock Center of Kyushu University (Fukuoka, Japan). The W chromosome of this line carries a chromosome fragment from chromosome 2 containing the pSa mutation, which changes the color of the larval epidermis from white to dark, allowing gynandromorphic larvae to be distinguished visually according to the color of the larval epidermis (Figure 1A). The standard silkworm strain p50T was also provided by the Silkworm Stock Center of Kyushu University. Gynandromorphic animals used in this study were obtained from eggs produced by crosses between m042 females and p50T males. Non-gynandromorphic control animals with the same genetic background as the gynandromorphic animals were obtained from eggs produced by crosses between p50T females and m042 females. The silkworm larvae were reared under standard conditions, as described previously [54].

### 2.2. RNA Extraction and Reverse Transcription (RT)-PCR

The total RNA was extracted from the fat bodies and gonads of the fifth instar larvae using ISOGEN (Nippon Gene, Tokyo, Japan), according to the manufacturer’s instructions. Reverse transcription and subsequent PCR were performed, as described previously [55]. Primer sequences used in the RT-PCR are shown in Appendix A.

### 2.3. Quantitative Real-Time RT-PCR (RT-qPCR)

A quantitative real-time RT-PCR (RT-qPCR) was performed according to the protocol described previously [55]. The amplification of *Bombyx mori elongation factor-2* (*BmEF-2*) was used as an internal standard reaction. The threshold cycle (CT) value was normalized relative to that of *BmEF-2* using the Multiple RQ software Ver 5.1.1 (TaKaRa Bio Inc., Kyoto, Japan). The primer sequences used in the RT-qPCR are described in Appendix A. The primer sets listed in Appendix A did not yield off-target products, as confirmed by dissociation curves of the resulting RT-qPCR products. The expression of each gene of interest relative to *BmEF-2* was determined in quadruplicate.

### 2.4. RNA-Seq Analysis

The total RNAs were extracted from the fat bodies of day 3 fifth instar larvae of gynandromorphic individuals and control males and females. The extracted RNAs were treated with TURBO™ DNase (Thermo Fisher Scientific, Waltham, MA, USA) to remove contaminated genomic DNA, according to the manufacturer’s instructions. The resulting total RNAs were subjected to RNA sequencing (RNA-seq) analysis by Eurofins Genomics (https://eurofinsgenomics.jp/jp/service/ngs/denovo_transcriptome/ (accessed on 21 July 2023)). cDNA libraries were prepared using a NovaSeq6000 S4 Reagent Kit and sequenced according to the standard protocol for paired-end analysis (read length: 150 bp/read) using NovaSeq6000. Approximately 54–60G reads were obtained from each sample. Analysis of differentially expressed genes (DEGs) between males and females was performed by Eurofins Genomics. Briefly, adapter sequences and low-quality reads were removed from the acquired reads using trimmomatic (Ver. 0.39), and the resulting reads were mapped onto the reference sequence using the BWA software (Ver. 0.7.17). Since publicly available RefSeq data in *B. mori* were established based on the data obtained from the male genome and thus did not contain sequences of female-specific genes such as *Fem* derived from the W chromosome, we constructed a reference sequence by integrating the RefSeq of *B. mori* (https://ftp.ncbi.nlm.nih.gov/genomes/all/GCF/014/905/235/GCF_014905235.1_Bmori_2016v1.0/GCF_014905235.1_Bmori_2016v1.0_rna.fna.gz (accessed on 28 July 2023)) and the *Fem* RNA sequence (NCBI accession number AB840787.1). To directly compare the expression level of each gene between different samples, the obtained data were normalized by the Trimmed Mean of M-values (TMM) method using edgeR (ver. 23.16.5).

### 2.5. Fat Body Culture

The fat bodies and hemolymph were obtained from day 3 fifth instar larvae of the p50 line. The fat bodies were incubated in hemolymph from the indicated sex for 1–2 h at room temperature. To prevent melanization, a 1/10 volume of 5% sodium thiosulfate was added to the hemolymph. The fat bodies cultured in an insect medium (TC-100; Invitrogen, Carlsbad, CA, USA) were used as negative controls. The extraction of the total RNA from the fat bodies after culturing was performed according to the method described above. The resulting total RNA was subjected to transcriptomics and RT-qPCR analyses, according to the protocol described above.

### 2.6. Statistics

All statistical analyses were performed using Microsoft Excel (Microsoft, Redmond, WA, USA) or Easy R (EZR) software Ver 1.60 (https://www.jichi.ac.jp/saitama-sct/SaitamaHP.files/download.html, accessed on 6 September 2021). A Shapiro–Wilk test was used to evaluate whether the data obtained in each experiment showed a normal distribution. As the number of samples was less than 25 and did not show a normal distribution, the significance of differences between the two groups was examined using a Mann–Whitney U test. Correlations between the two groups were verified by calculating Pearson’s correlation coefficient (*r*). To determine whether there was a significant linear correlation between the two groups, we tested the significance of the correlation coefficient by calculating the *p*-value.

## 3. Results

### 3.1. The Sex Differentiation Status in the Fat Body of Gynandromorphic Larvae

The silkworm strain m042, which frequently produces gynandromorphic individuals, was used in this study (see Section 2). This line allows the selection of gynandromorphic animals based on the color of the larval epidermis, i.e., dark areas are genetically female, and white areas are genetically male (Figure 1A). First, we investigated the status of sexual differentiation in the fat body of the selected gynandromorphic larvae. Five larvae with a symmetrically different epidermis color at the midline of the body were selected as gynandromorphic animals (Figure 1B), and the total RNA was extracted independently from the right and left sides of their fat bodies. RT-qPCR analysis showed that the expression level of *Fem*, a master regulatory gene for femaleness, was higher in the right side than the left side of the fat body in gynandromorphic larvae #1, #3, and #5 (Figure 1B). On the other hand, the expression level of *ImpM*, which has been used as a male marker in previous studies [45,46,47,48], was greater in the left than the right side of the fat body in larvae #2–#5 (Figure 1B). RT-PCR analysis indicated that the sex-specific splicing pattern of *Bmdsx* pre-mRNA was different between the left and right sides of the fat body in larvae #4 and #5 (Figure 1B). In addition, both male and female modes of *Bmdsx* expression were observed in both sides of the fat body in larva #2. Ten larvae, each of which was either dark or white in color throughout the body, were subjected to the same RT-PCR analysis. All dark-colored larvae showed expression of the female type of *Bmdsx* (*BmdsxF*), while only the male mode of *Bmdsx* (*BmdsxM*) expression was observed in all the white larvae (Appendix A). These results suggested that the sex of the epidermis does not accurately reflect the sex of the fat body, but that the sex differentiation status in the fat body of the larvae selected as gynandromorphic based on the color of the epidermis differs between distinct areas of the fat body or shows intersexuality. 

### 3.2. The Gynandromorphic Status of the Fat Body in Gynandromorphic Silkworms

To verify the gynandromorphic status of the fat body, we quantitatively compared the expression level of the W-linked gene *Fem* between different areas of the fat body obtained from gynandromorphic day 3 fifth instar larvae. The total RNA was extracted from four different areas of the fat body (LA, left anterior; LP, left posterior; RA, right anterior; and RP, right posterior) and examined through RT-qPCR (Figure 2A). Gynandromorphic larvae exhibiting a dark epidermis on the right half of the body (i.e., ZW) and a white epidermis on the left half of the body (i.e., ZZ) were used in the analysis (Figure 2B). The female larva has a pair of ovaries that are nearly triangular in shape, while the male larva has a pair of testes, each consisting of four lobes (Figure 2B). The gonadal sex in the gynandromorphic larvae was consistent with that expected from the epidermis color, i.e., they had a pair of gonads consisting of an ovary on the right side and a testis on the left side (Figure 2B). The *Fem* expression level in the control females was almost equally high in all the areas examined (Figure 2C). There were no significant differences in the expression of *Fem* between the right and left sides of the fat bodies (Figure 2D). The same RT-qPCR analysis demonstrated that the expression level of *Fem* in the gynandromorphic animals was markedly different among the four areas, ranging from almost zero to a level comparable to the control females (Figure 2C). Consistent with the sex expected from epidermis color, the expression level of *Fem* was significantly higher in the right than in the left side of the fat body (Figure 2D). These results suggest that the level of *Fem* expression in the gynandromorphic silkworms is regulated cell-autonomously depending on the genetic sex of each cell. Next, we investigated the expression level of the storage protein 1 gene (*sp-1*), as its expression in the fat body is known to be regulated in a genetic sex-dependent manner (i.e., the presence or absence of the W chromosome) [56,57]. SP-1 is a plasma protein that is synthesized exclusively in the fat body of the female silkworm [56,58,59]. Similar to *Fem* expression, control females expressed *sp-1* at almost equal levels in all areas of the fat body examined (Figure 2C). There were no significant differences in *sp-1* expression between the right and left sides of the fat bodies (Figure 2D). On the other hand, the level of *sp-1* expression in the gynandromorphic larvae differed markedly among the four areas in relation to the level of *Fem* expression in the corresponding areas (Figure 2C). Differences in the expression level that were 37–230-fold were observed within the same individual. The expression of *sp-1* was also significantly different between the right and left sides of the fat body (Figure 2D). Furthermore, there was a strong correlation (*r* = 0.907, *p* = 9.8 × 10^–10^) between the levels of *Fem* and *sp-1* expression in each area (Figure 2E). These results strongly suggested that the fat bodies derived from gynandromorphic individuals used in this study were composed of ZZ and ZW cells. 

### 3.3. Transcriptome Analysis of the Fat Body in Gynandromorphic Silkworms

To investigate whether the gynandromorphic status affects sexual dimorphism traits of the fat body at the level of the transcriptome, we performed RNA-seq analysis using the total RNAs extracted from the fat bodies of day 3 fifth instar gynandromorphic larvae. RT-PCR analysis showed that the gynandromorphic fat body expressed both male and female types of *Bmdsx* (Appendix A, mos1 and mos2). On the other hand, the sex-specific expression of *Bmdsx* was observed in the control males and females (Appendix A). Comparative analysis of the transcriptome between control males and females successfully identified 242 sex-differentially expressed genes (Figure 3A). Hierarchical clustering analysis focusing on the levels of the sex-differentially expressed genes suggested that the sexual differentiation status in the fat body of the gynandromorphic individuals was different from both males and females (Figure 3B). To further select genes showing more reliable sex differences, genes with normalized TMM values of less than 10 were excluded from the identified 242 genes. We designated the 232 genes selected by this treatment as S-DEGs (sex-differentially expressed genes) (Appendix A) and used perturbations in the expression levels of S-DEGs as a molecular index to estimate the degree of sexual differentiation of the fat body. 

If the S-DEG expression level is regulated in a cell-autonomous manner, the level should be correlated with the proportion of male cells (ZZ cells) and female cells (ZW cells) constituting the fat body. To examine whether this hypothesis could explain the unique expression pattern of S-DEGs observed in the gynandromorphic fat body, we estimated the proportions of female and male cells in the gynandromorphic fat body based on the expression level of the W-linked female determining gene, *Fem* [45]. RNA-seq data showed that the expression levels of *Fem* in the two gynandromorphic animals (mos1 and mos2) were less than half that in control females (Figure 3C, upper panel). The validity of the *Fem* expression level obtained by RNA-seq was confirmed by RT-qPCR analysis (Figure 3C, upper panel). The population of ZW cells in the gynandromorphic fat body was calculated based on the expression level of *Fem* in control females, i.e., if the Fem expression level was equivalent to that of the control females, all the cells of the fat body were considered to be ZW cells, while if the Fem expression level was half that in females, 50% of the fat body cells were considered to be ZW cells. The proportion of ZW cells was estimated to be 33.4% for mos1 and 46.1% for mos2 (Figure 3C, lower panel).

We next predicted the expression levels of S-DEGs in mos1 and mos2 based on the acquired population of ZZ/ZW cells. If the expression level of S-DEGs in the gynandromorphic fat body is regulated in a cell-autonomous manner, the expression level can be predicted by the following formula: expression level of one S-DEG in gynandromorphic fat body = (expression level of the S-DEG in control female × population of ZW cells) + (expression level of the S-DEG in control male × population of ZZ cells). The degree of coincidence between the expression levels predicted by this formula and the expression levels of S-DEGs determined by RNA-seq was investigated according to the method described in Figure 3D. The results showed that the predicted expression levels of 207 of the 232 S-DEGs were almost consistent with the expression levels determined by RNA-seq (Figure 3D). Moreover, the expression level of each S-DEG predicted by the above formula was strongly correlated with the expression level in the fat body of both mos1 and mos2 determined by RNA-seq analysis (Figure 3E, *r* = 0.984 in mos1, *r* = 0.996 in mos2). These results strongly support the hypothesis that the expression levels of most S-DEGs in the fat body of gynandromorphic animals are regulated in a cell-autonomous manner.

### 3.4. Identification of S-DEG Expression Which Is Regulated Non-Cell-Autonomously

As mentioned above, several previous studies indicated the existence of genes whose sexually dimorphic expression is stimulated by of hormones such as ecdysone [31,32,33,34,35,37,38,39]. To identify such non-cell-autonomously regulated genes, we searched for S-DEGs whose expression in our RNA-seq samples differed significantly from the predicted values. For this purpose, we extracted S-DEGs plotted outside the interquartile range in the box-and-whisker diagram illustrated in Figure 3D. This analysis suggested that the expression levels of 25 S-DEGs deviated significantly from the predicted values (highlighted in yellow in Appendix A). Among these S-DEGs, we focused on the vitellogenin gene (*BmVg*) and the basic juvenile hormone-suppressible protein 2 gene (*JhSP-2*), since the transcription of these two genes is known to be affected by the presence of hormones, such as 20E and the juvenile hormone (JH) [60,61]. Similar to *sp-1*, these two genes are expressed exclusively in the female fat body at the fifth instar larval stage [60,61]. Consistent with previous reports, our RNA-seq data also showed female-biased expression of *BmVg* and *JhSP-2*, the expression levels of which were 10-fold and 100-fold higher in females than males, respectively (Appendix A).

To examine whether the expression levels of these two genes in different areas of the fat body are correlated with the levels of Fem expression in the corresponding areas, RT-qPCR analysis was performed as described above. Both *BmVg* and *JhSP-2* showed different expression levels among the four areas examined in gynandromorphic animals, as observed for *sp-1* (Figure 4A). However, the levels of *BmVg* expression in the right and left sides of the fat body showed no significant differences in the gynandromorphic animals, even though the levels of *Fem* expression were significantly different between the two sides (Figure 4B). A comparison between anterior and posterior areas of the fat body showed significant differences in the level of *BmVg* expression in the control females, although the same comparison showed no significant spatial differences in *Fem* expression (Figure 4B). Consistent with these observations, the correlations between the expression levels of *BmVg* and *JhSP-2* and that of *Fem* were lower than that in the case of *sp-1* (Figure 4C). These results support the possibility that the sexually dimorphic expression of *BmVg* and *JhSP-2* may be regulated by other factors, in addition to *Fem*.

### 3.5. Effects of Hemolymph on Sexually Dimorphic Expression of S-DEGs

To examine whether sexually dimorphic *BmVg* and *JhSP-2* expression is regulated by external factors such as hormones, we conducted an in vitro culture of the fat body using hemolymph collected from day 3 fifth instar larvae, followed by RT-qPCR quantification of the expression levels of the above two genes. Fat bodies without culture and fat bodies cultured with insect culture medium (TC-100) were used as positive and negative controls, respectively (Figure 5, w/o culture and TC-100). In vitro culture for 1 h did not affect the level of *sp-1* expression under any conditions (Figure 5A); the *sp-1* expression level was strongly correlated with that of *Fem* (Figure 2B,C). This is consistent with a previous report which showed that *sp-1* expression is regulated in a genetic sex-dependent manner [56,57]. On the other hand, both *BmVg* and *JhSP-2* expression levels in female fat bodies were significantly decreased when cultured with insect medium or male hemolymph in comparison to the positive controls (Figure 5B,C). The expression level of *JhSP-2* in male fat bodies was significantly increased when cultured in female hemolymph (Figure 5C). A similar increasing trend was observed for the expression level of *BmVg* in male fat bodies cultured with female hemolymph (Figure 5B). These results suggested that the female hemolymph contains factors that positively regulate the expression levels of *BmVg* and *JhSP-2* in the fat body.

## 4. Discussion

It has been long believed that insects do not have sex hormones and that their sex is determined in a cell-autonomous manner. However, there is accumulating evidence that sexual differentiation in insects is regulated by hormonal factors, as seen when focusing on secondary sexual traits. This study was performed to quantify the integrity of secondary sexual trait development in gynandromorphic silkworms to clarify the contributions of cell-autonomous and non-cell-autonomous regulation. As a secondary sexual trait, we focused on the fat body because it shows remarkable sexual dimorphism in physiological properties represented by the biosynthesis of plasma proteins and proteins involved in oogenesis regulated by both cell-autonomous and non-cell-autonomous mechanisms [37,56,58,59,60,62,63].

In this study, gynandromorphic silkworms were selected according to the color of the larval epidermis, i.e., dark color indicates genetically female silkworms, and white indicates genetically male silkworms (Figure 1A). Five larvae with a symmetrically different epidermis color at the midline of the body were used in the analyses (Figure 1B). The expression of sex-determining genes in the fat bodies of two gynandromorphic silkworms did not reflect the sex predicted from the epidermis color (Figure 1B, #1, and #3). This was likely due to the differences in the germ layers from which these two tissues originate. The epidermis is derived from the ectoderm, whereas the fat body originates from the mesoderm in silkworms [64]. The mesoderm is retracted from the thorax to the abdomen and is divided into segments around 36 h after fertilization. Subsequently, a neural groove derived from the ectoderm is formed, and its invagination into the coelom splits the mesoderm into right and left lobes [64]. This invagination of the ectoderm causes the symmetrical differences in color of the epidermis observed in the gynandromorphic silkworms (Figure 1). Different degrees of sex mosaicism in the cell populations comprising the ectoderm and mesoderm may lead to discrepancies between the sex predicted from the color of the epidermis and the expression pattern of sex-determining genes in the fat body.

In this study, we quantified the expression levels of *Fem* and *sp-1* at several different sites in the fat body (Figure 2). Our results showed that *Fem* expression is regulated cell-autonomously and that the levels of *sp-1* and *Fem* expression are strongly correlated, suggesting that the fat body of the gynandromorphic silkworm is composed of ZZ and ZW cells. A similar gynandromorphic status of the fat body obtained from a genetic mosaic silkworm strain was also reported previously [56]. These findings enabled us to estimate the populations of ZZ and ZW cells of the gynandromorphic fat bodies based on the level of Fem expression (Figure 3C).

To investigate whether the gynandromorphic status affects sexual dimorphism traits of the fat body at the level of the transcriptome, we performed RNA-seq analysis using the total RNAs extracted from the fat bodies, and successfully identified 232 S-DEGs (Appendix A). A total of 50 of these 242 S-DEGs were uncharacterized genes. Among them, 11 genes encoded non-coding RNAs (ncRNAs). A blastn search using each ncRNA sequence as a query sequence revealed that nine of them were *Bombyx mori* specific and that the remaining two (LOC119630117 and LOC119629108) were widely conserved among lepidopteran species. More importantly, LOC119630117 showed male-specific expression, while LOC119629108 was expressed only in females (Appendix A). These results indicate that these two ncRNAs might be relevant to the sex differentiation process unique to lepidopteran insects.

Sex differences in body size, tolerance to disease, and longevity have been reported in various organisms, including the silkworm. Interestingly, all antibacterial peptide genes in S-DEGs (cecropin-B, cecropin B2, cecropin-D-like peptide, gloverin 1, gloverin 4, gloverin 4-like, gloverin-like, and antibacterial peptide enbocin 2) were more highly expressed in males than in females (Appendix A). In contrast, in females, all genes involved in protein biosynthesis in S-DEGs (eukaryotic initiation factor 4E-2, ribosome biogenesis regulatory protein homolog, ribosomal protein L7Ae, ribosome biogenesis protein BRX1 homolog, rRNA processing protein Ebp2, etc.) were highly expressed as compared with males. In addition, short neuropeptide F and its putative receptor (neuropeptide receptor B1), both of which are closely related to promote food intake behaviors [65], were more highly expressed in females than in males (Appendix A). Sex differences in the expression levels of these genes in fat bodies might govern sexual differences in body size and tolerance to disease.

The most striking finding of this study was that the expression levels of almost all the genes showing sexually dimorphic expression in the fat body varied, according to the precise proportion of ZZ and ZW cells (Figure 3D,E). That is, the sexual dimorphism of the transcriptome in the fat body is regulated cell-autonomously under the control of a single W-linked gene, *Fem*. This study provided strong evidence that external factors, such as hormones, are dispensable for the development of sexually dimorphic traits, including secondary sexual traits, in insects.

On the other hand, our analysis supported the possibility that the sexually dimorphic expression of 25 of the 232 S-DEGs is regulated in a non-cell-autonomous manner (Figure 3D, Figure 4 and Appendix A). In fact, in vitro culture experiments demonstrated that the female-biased expression of some of these genes (*BmVg* and *JhSP-2*) was partially regulated by an extracellular factor(s) specifically present in the female hemolymph (Figure 5B,C). A previous study showed that the transcription of *JhSP*-2 is repressed by treatment with a JH analog [60]. In lepidopteran insects, Vg synthesis is stimulated by 20E, and the topical application of JH is known to inhibit *Vg* gene expression [65,66]. These observations support the suggestion that the lower JH titer in the female compared to the male hemolymph resulted in significant increases in *BmVg* and *JhSP-2* expression levels in the fat bodies cultured in the female hemolymph. However, to our knowledge, there have been no reports of sex-related differences in the JH titer in the silkworm larval hemolymph. In the tobacco hawk worm *Manduca sexta*, JH production stops in males but not in females in the last larval instar, resulting in a sex-related difference in the JH titer [67]. In contrast, Baker et al. (1987) detected no sex-specific differences in the degree or timing of JH, JH esterase, and ecdysteroid titers in *M. sexta* [68]. It seems unlikely that a sex-related difference in the JH titer is a key factor in the female-biased expression of *BmVg* or *JhSP-2*. As in many vertebrate species, some sex-specific hormonal factors secreted from the gonads may control the expression of these genes. The transplantation of the ovary into males was reported to increase *BmVg* synthesis in the fat bodies of male silkworms [61]. These authors also showed that the injection of estradiol (17-beta estradiol; E2) into silkworm larvae can induce *BmVg* synthesis in male fat bodies [61]. As E2 is synthesized in the silkworm ovary and the E2-binding protein (estrogen receptor; ER) is localized to the nucleus of the fat body [69,70], it is possible that ovary-derived E2 may contribute to the regulation of the female-specific expression of a subset of S-DEGs in the fat body.

It should also be noted that the expression levels of *BmVg* and *JhSP-2* between four different sites in the fat body differed not only in gynandromorphic animals but also in the control females (Figure 4A). The level of *BmVg* expression was significantly higher in the posterior area than the anterior area in the control female fat bodies, although there were no significant differences in the Fem expression level (Figure 4B). These results suggest that *BmVg* and *JhSP-2* may exhibit site-specific expression patterns in the fat body. Previous studies showed that the fat body is heterogeneous and has functional differences from site to site [71]. The fat body is divided into the peripheral fat body lining the epidermis and the visceral fat body adjacent to the gut. The peripheral fat body actively synthesizes proteins, whereas the visceral fat body is specialized for the storage of proteins and lipids [71,72]. The storage and secretory activity of storage proteins, such as SP-1 and SP-2, in the peripheral fat body decreases from the posterior to dorsal and abdominal regions [73]. The amount of BmVg storage is highest in the abdominal area of the peripheral fat body, while there is little BmVg storage in the fat body around the hindgut [73]. These observations may explain why the expression levels of *BmVg* and *JhSP-2* varied among areas in the fat body examined in this study.

The expression levels of *JhSP-2* and *BmVg* showed weak or moderate correlations with those of *Fem* (Figure 4C), suggesting that they are regulated, to some extent, according to the sex of each cell. In addition, the expression of *JhSP-2* and *BmVg* may be regulated by extracellular factors secreted from tissues other than the fat body, such as the gonads, and may also be regulated in a site-dependent manner. It is likely that a combination of these factors determines the expression levels of *BmVg* and *JhSP-2*. Similar explanations may be applicable to the remaining 23 S-DEGs that were also predicted to be regulated in a non-cell-autonomous manner (Figure 3D, Figure 4 and Appendix A).

## 5. Conclusions

In this study, we analyzed the dynamics of the sexually dimorphic transcriptome in gynandromorphic individuals of the hereditary mosaic mutant strain in the silkworm to quantitatively evaluate the degrees of cell-autonomous and non–cell-autonomous regulation of secondary sexual trait development. As a result, 232 S-DEGs were successfully identified by comparing the transcriptomes between male and female fat bodies. Strikingly, the expression levels of 217 of the 232 S-DEGs varied precisely according to the proportion of ZZ and ZW cells in the gynandromorphic silkworms. Our results also suggested that the female-biased expression of several genes, such as those encoding vitellogenin and plasma proteins, in the fat body may be regulated by extracellular factors in the hemolymph, but their contributions to sexual differentiation are limited. Thus, when sexual dimorphism is viewed at the transcriptome level, we concluded that secondary sexual characteristics in the fat body are controlled in a cell-autonomous manner. However, it is premature to completely exclude the involvement of extracellular factors, such as hormones, in the development of sexually dimorphic traits in the fat body, as the roles of JH and ecdysone in the yolk protein and vitellogenin synthesis vary widely among insect species [74]. In addition, it remains to be verified whether the same is true for tissues other than the fat body. The application of the approach used in this study to various tissues may provide insight into whether insect sexual differentiation is indeed regulated cell-autonomously.

## Figures and Tables

**Figure 1 jdb-12-00031-f001:**
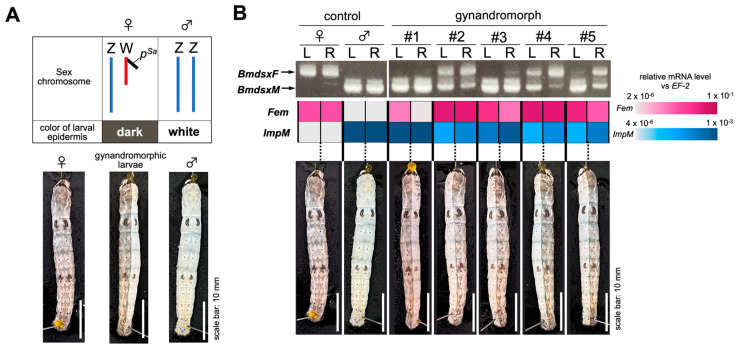
The silkworm strain m042 carrying the mo (hereditary mosaic) mutations examined in this study. (**A**) The upper panel shows the relation between sex chromosome composition and the color of the larval epidermis in the m042 strain. Blue vertical lines show the Z chromosomes and the red vertical line indicates the W chromosome. The W chromosome in this line carries a fragment from chromosome 2 (indicated by the black line) containing the pSa mutation, which changes the color of the larval epidermis from white to dark. The lower panel shows day 3 fifth instar larvae of the m042 strain. Female larvae exhibit a dark epidermis due to the W−linked pSa mutation. The epidermis of gynandromorphic larvae composed of both ZW and ZZ cells is both dark and white in color. (**B**) Status of sexual differentiation in the fat body of the gynandromorphic larvae. The total RNA was extracted independently from the right and left sides of the fat bodies and subjected to expression analysis of genes involved in the sex determination and sexual differentiation of the silkworm. The upper panel shows the expression patterns of *Bmdsx*. RT−PCR products were separated by electrophoresis through 1.5% agarose gels containing ethidium bromide (1 mg/mL). The numbers #1–#5 indicate the individual gynandromorphic larvae; R and L indicate the right and left sides of the fat body, respectively. The arrows to the left of the gel refer to the positions of *BmdsxF* and *BmdsxM*. The middle panel shows the expression levels of *Fem* and *ImpM* quantified by RT−qPCR. The expression level of each gene is color−coded with reference to the indicators shown on the right side of the figure. The lower panel shows a photograph of the day 3 fifth instar larvae used in the analysis.

**Figure 2 jdb-12-00031-f002:**
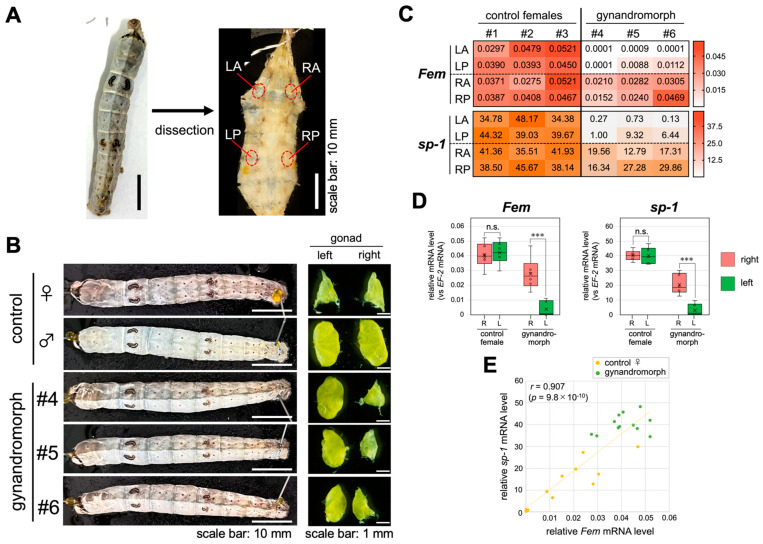
The gynandromorphic status of the fat body in the gynandromorphic silkworm. (**A**) The total RNA was extracted from four different areas of the fat body (LA, left anterior; LP, left posterior; RA, right anterior; and RP, right posterior) and examined through RT-qPCR analysis. (**B**) The left panel shows the dorsal view of the control day 3 fifth instar male and female and gynandromorphic larvae. Three gynandromorphic larvae (#4–#6) were subjected to analysis. The right panel shows a pair of gonads dissected out from the larvae indicated in the left panel. (**C**) The expression levels of *Fem* and *sp-1* in each area of three control females and three gynandromorphic individuals were quantified through RT-qPCR. All values are shown relative to the expression of *EF-2*. The expression level is also indicated by color according to the scale shown in the box to the right of each table. The numbers #1 to #6 in the table indicate individual larvae. (**D**) The expression levels of *Fem* and *sp-1* were compared between the right and left of the fat bodies of control females and the gynandromorphic larvae. The values obtained from each sample and their distributions are represented by box-and-whisker plots. Error bars indicate the standard deviation. R and L indicate right and left, respectively. n.s., not significant. *** *p* < 0.01 (Mann–Whitney U test). (**E**) A scatter plot of the expression levels of *Fem* and *sp-1* in each area of the fat bodies, as described in (**C**). The vertical axis indicates the expression level of *sp-1* and the horizontal axis indicates the expression level of *Fem*. *r*, Pearson’s product–rate correlation coefficient; *p*-values are for the no-correlation test.

**Figure 3 jdb-12-00031-f003:**
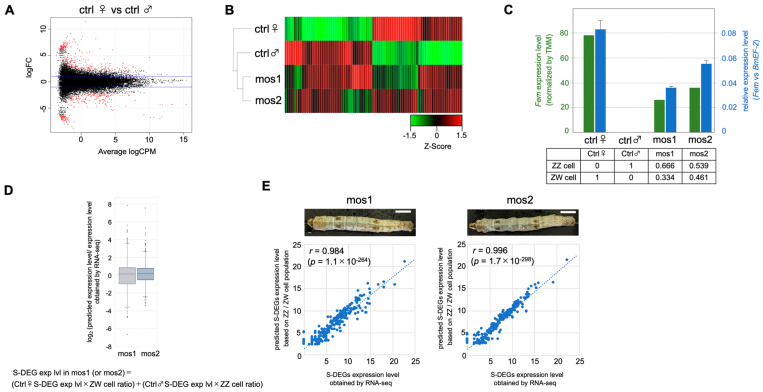
Analysis of the transcriptome in the fat bodies of gynandromorphic silkworms. (**A**) An MA−plot between control males and control females. The total of 242 sex−differentially expressed genes, i.e., genes with significant differential expression between males and females (FDR ≤ 0.05), are shown in red. (**B**) A heat map and hierarchical cluster of the sex−differentially expressed genes among the control males, control females, and gynandromorphic animals (mos1 and mos2). (**C**) The expression levels of *Fem* in each sample estimated by RNA−seq analysis are indicated by the green columns, and the relative expression level of *Fem* quantified by RT-qPCR is shown by the blue columns. The populations of ZZ and ZW cells in the fat bodies of the control males, control females, and gynandromorphic animals (mos1 and mos2) were calculated based on the expression level of *Fem* and are shown in the table. (**D**) The expression levels of each S−DEG in the gynandromorphic animals (mos1 or mos2) were predicted by the following formula: expression level of S−DEG in mos1 (or mos2) = (expression level of the S-DEG in control female × ratio of ZW cells) + (expression level of the S−DEG in control male × population of ZZ cells). The degrees of coincidence between the predicted expression levels and the levels of S−DEG expression determined by RNA−seq in the gynandromorphic animals (mos1 and mos2) were estimated by log_2_ (predicted expression level/expression level of S−DEG determined by RNA-seq). The estimated values and their distributions are represented by box−and−whisker plots. Dots show outliers. (**E**) A scatter plot of the predicted expression levels of S−DEGs and those determined by RNA−seq in mos1 and mos2. Scale bar: 10 mm. The vertical axis indicates the predicted expression levels and the horizontal axis indicates the expression level determined by RNA−seq. *r*, Pearson’s product–rate correlation coefficient; *p*−values are for the no−correlation test.

**Figure 4 jdb-12-00031-f004:**
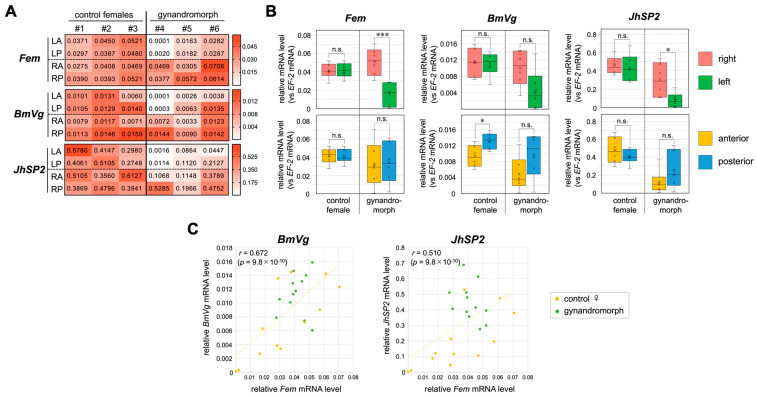
The expression levels of *BmVg* and *JhSP-2* in different areas of the fat body and their correlations with *Fem* expression levels. (**A**) The expression levels of *Fem*, *BmVg*, and *JhSP-2* in each area of three control females and three gynandromorphic individuals were quantified by RT−qPCR. Values are shown relative to the level of *EF-2* expression. The expression level is also indicated by color according to the scale shown in the box to the right of each table. In the table, #1–#6 indicate individual larvae. (**B**) The expression levels of *Fem*, *BmVg*, and *JhSP-2* were compared between the right and left areas of the fat bodies of control females and gynandromorphic larvae (upper panel). The same comparison was performed between the anterior and posterior parts of the fat bodies (lower panel). The values obtained from each sample and their distributions are represented by box−and−whisker plots. Error bars indicate the standard deviation. R and L indicate right and left, respectively. A and P, anterior and posterior, respectively. n.s., not significant. * *p* < 0.05, *** *p* < 0.01 (Mann–Whitney U test). (**C**) Scatter plots of the expression levels of *Fem* and *BmVg* (left panel) and *Fem* and *JhSP-2* (right panel) in each area of the fat bodies described in (**A**). The vertical axis indicates the expression levels of *BmVg* or *JhSP-2*. The horizontal axis indicates the expression level of *Fem*. *r*, Pearson’s product–rate correlation coefficient; *p*−values are for the no−correlation test.

**Figure 5 jdb-12-00031-f005:**
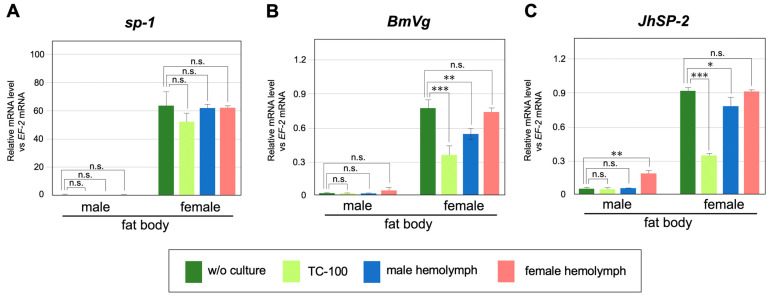
The effects of hemolymph on the sexually dimorphic expression of *sp-1*, *BmVg*, and *JhSP-2*. Male and female fat bodies were cultured for 1 h in hemolymph collected from male or female day 3 fifth instar larvae. The expression levels of (**A**) *sp-1*, (**B**) *BmVg*, and (**C**) *JhSP-2* in the cultured fat bodies were quantified by RT-qPCR. w/o culture, without culture. TC-100 is a culture medium specifically for insect cells. The fat bodies were collected from four male or female larvae and split into four experimental groups (w/o culture, TC-100, male hemolymph, and female hemolymph). Values represent the means ± SE of quadruplicate qPCR reactions for each sample from one representative of three independent experiments. n.s., not significant, * *p* < 0.05, ** *p* < 0.03, *** *p* < 0.01 (Mann–Whitney U test).

## Data Availability

The silkworm strains used in this study are reared continuously and passaged at the Laboratory of Bioresource Regulation, Department of Integrated Biosciences, Graduate School of Frontier Sciences, The University of Tokyo, and are available from the Institute of Genetic Resources, Faculty of Agriculture, Kyushu University. All raw sequence data obtained by RNA-seq analysis were deposited in the NCBI Sequence Read Archive (SRA) under BioProject accession number PRJDB18820, BioSample accession number SAMD00820453-SAMD00820456, Experiment accession number DRX577230-DRX577233, and Run accession number DRR596683-DRR596686. All data described in this study are available upon request.

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
