# Peer review of "Transcriptomic Evidence for Cell-Autonomous Sex Differentiation of the Gynandromorphic Fat Body in the Silkworm, *Bombyx mori"

_jdb, 2024, doi:10.3390/jdb12040031_

Round 1
Reviewer 1 Report
Comments and Suggestions for Authors
The authors present a clear and logical case ruling out humoral, non-autonomous regulation of most sexually dimorphic genes in the Bombyx fat body. Interestingly, about 10% of the dimorphic genes do appear to incorporate non-autonomous influences into their expression regulation, consistent with known sexually dimorphic production of hormones such as ecdysone.
The results appear solid. The only real way to improve the methodology might be to perform single-cell or single-nuclear RNA-seq to very rigorously nail down the correlation between, e.g., Fem and other genes, but that would be an expensive exercise that likely wouldn't change the overall message.
The figures are clear, and the text is written well. I have two small modifications in that regard:
Rearrange the sentence in lines 379-381 to make it more grammatical: “To identify such non-cell-autonomously regulated genes, we searched for S-DEGs whose expression in our RNA-seq samples differed significantly from the predicted values.”
The sentence in line 501 seems to have lost its verb – “differed not only ..."
An interesting but optional comparison would be to consult other transcriptional atlases, such as the Drosophila resource FlyAtlas, to assess whether these transcriptional differences are conserved. Of course FlyAtlas did not perform any sex-specific larval assays, but adult data is reported by sex. (This adds a second difference between your data and theirs -- taxon and developmental stage -- but perhaps something interesting will appear in any case.)
A further note on the resulting data set: there are very many "uncharacterized" genes in your list (S3). Even though they haven't been characterized, do these genes have a predicted function, or do they have conserved orthologues whose function is unknown, or are these Bombyx-specific (et c)? Are there any trends or common themes among the non-autonomously regulated genes? Are there any dimorphisms -- morphological, functional, behavioral, et c. -- in the Bombyx larva that might result from the ~250 transcriptional dimorphisms (you don't have to find them, just mention or speculate)?
Comments on the Quality of English LanguageThe text is written well. I have two small modifications:
Rearrange the sentence in lines 379-381 to make it more grammatical: “To identify such non-cell-autonomously regulated genes, we searched for S-DEGs whose expression in our RNA-seq samples differed significantly from the predicted values.”
The sentence in line 501 seems to have lost its verb – “differed not only ..."
Author Response
We appreciate the many valuable comments. We have responded to each and every comment with the utmost sincerity. We hope that our response will satisfy the reviewer. Our alterations as a result of the reviewer’s comments are as follows.
The authors present a clear and logical case ruling out humoral, non-autonomous regulation of most sexually dimorphic genes in the Bombyx fat body. Interestingly, about 10% of the dimorphic genes do appear to incorporate non-autonomous influences into their expression regulation, consistent with known sexually dimorphic production of hormones such as ecdysone.
The results appear solid. The only real way to improve the methodology might be to perform single-cell or single-nuclear RNA-seq to very rigorously nail down the correlation between, e.g., Fem and other genes, but that would be an expensive exercise that likely wouldn't change the overall message.
The figures are clear, and the text is written well. I have two small modifications in that regard:
(1) Rearrange the sentence in lines 379-381 to make it more grammatical: “To identify such non-cell-autonomously regulated genes, we searched for S-DEGs whose expression in our RNA-seq samples differed significantly from the predicted values.”
Response: We rearranged the sentence according to the reviewer's comment. Pleas also see lines 385-387 in the revised manuscript.
(2) The sentence in line 501 seems to have lost its verb – “differed not only ..."
Response: Sorry for missing the verb. We inserted "differed" in the sentence. Pleas also see line 530 in the revised manuscript.
(3) An interesting but optional comparison would be to consult other transcriptional atlases, such as the Drosophila resource FlyAtlas, to assess whether these transcriptional differences are conserved. Of course FlyAtlas did not perform any sex-specific larval assays, but adult data is reported by sex. (This adds a second difference between your data and theirs -- taxon and developmental stage -- but perhaps something interesting will appear in any case.)
Response: We appreciate the interesting suggestions proposed by the reviewer. However, as the reviewer himself pointed out, comparison on different developmental stages and different tissues would make interpretation of the results difficult. In addition, the main scope of this study is to verify the presence of cell-autonomous sexual differentiation. Accordingly, we would like to conduct a study based on this proposal by the reviewer at another opportunity.
(4) A further note on the resulting data set: there are very many "uncharacterized" genes in your list (S3). Even though they haven't been characterized, do these genes have a predicted function, or do they have conserved orthologues whose function is unknown, or are these Bombyx-specific (et c)? Are there any trends or common themes among the non-autonomously regulated genes? Are there any dimorphisms -- morphological, functional, behavioral, et c. -- in the Bombyx larva that might result from the ~250 transcriptional dimorphisms (you don't have to find them, just mention or speculate)?
Response: We thank the reviewer’s suggestions. As the reviewer pointed out, S-DEGs contained 50 uncharacterized genes. Based on the suggestions, we investigated a predicted function of each uncharacterized gene. Functional analysis using InterProScan (https://www.ebi.ac.uk/interpro/search/sequence/) identified a predicted function of 17 of 50. However, we could not find any trend or common themes among the non-autonomously regulated genes. On the other hand, we found that 11 of 50 uncharacterized genes encode non-coding RNA (ncRNA). blastn search using the ncRNA sequence as a query sequence demonstrated that nine of them were Bombyx mori specific and the remaining two (LOC119630117 and LOC119629108) were widely conserved among lepidopteran species. More importantly, LOC119630117 showed male-specific expression while LOC119629108 expressed only in females (Table S3). These results indicate that these two ncRNAs might be relevant to the sex differentiation process unique to lepidopteran insects.
Sex differences in body size, tolerance to disease, and longevity have been reported in various organisms including the silkworm. The fat body synthesizes a variety of physiologically active substances and serves as a center for metabolic control and immune response [50, 51]. Interestingly, all antibacterial peptide genes in S-DEGs (cecropin-B, cecropin B2, cecropin-D-like peptide, gloverin 1, gloverin 4, gloverin 4-like, gloverin-like, and antibacterial peptide enbocin 2) were highly expressed in males than in females (Table S3). In contrast, in females, all genes involved in protein biosynthesis in S-DEGs (eukaryotic initiation factor 4E-2, ribosome biogenesis regulatory protein homolog, ribosomal protein L7Ae, ribosome biogenesis protein BRX1 homolog, rRNA processing protein Ebp2, etc.) were highly expressed as compared with males. In addition, short neuropeptide F, and its putative receptor (neuropeptide receptor B1), both of which are closely related to promote food intake behaviors, were highly expressed in females than in males (Table S3). Sex differences in the expression levels of these genes in fat bodies may govern sexual differences in body size and tolerance to disease.
We added the above findings and discussions about the findings to Discussion section. Please also see lines 474-495.
Reviewer 2 Report
Comments and Suggestions for Authors
In this manuscript the authors use a gynandromorphic strain of Bombyx mori to investigate genetic differences between male and female cells of the fat body of larvae.
Transcriptomic analyses identify 242 DEG that are differentially expressed in this tissue between the sexes. In gynandromorphic larvae the variation of expression of these DEGs correlate with the percentage of male or female cells in the tissue, thus concluding that sex determination of secondary sexual traits is probably cell autonomous. However that does not rule out the possibility of hormonal involvment in tissue differentiation as well.
Overall, the manuscript is well designed and carefully check all the experimental problems that could arise.
I only have one concern in the analysis of RNAseq that I would like the authors to adress. The MA plot of Fig. 3A shows significant DEGs on the left side of the graph, identifying genes with low cpm. Typically, genes that are not highly expressed will be artefactually called DEGs in this portion of the graph. Normalization method often shrink this fraction of the transcriptome or even filter it out entirely to exclude low counts genes (see DESeq2 manual for example, even though the authors used edgeR). This is not specified in the methods.
Similarly, the alignment methods is not entirely described. I was concerned about the reference used to align. The authors mention (l. 175) the integration of two nucleotide sequences. Why ? What is the resulting reference file ? Can we have brief stats in a supplementary information on the number of genes in total, redundant or not, representing alternative splicing or not etc... ?
Author Response
We appreciate the many valuable comments. We have responded to each and every comment with the utmost sincerity. We hope that our response will satisfy the reviewer. Our alterations as a result of the reviewer’s comments are as follows.
In this manuscript the authors use a gynandromorphic strain of Bombyx mori to investigate genetic differences between male and female cells of the fat body of larvae.
Transcriptomic analyses identify 242 DEG that are differentially expressed in this tissue between the sexes. In gynandromorphic larvae the variation of expression of these DEGs correlate with the percentage of male or female cells in the tissue, thus concluding that sex determination of secondary sexual traits is probably cell autonomous. However that does not rule out the possibility of hormonal involvment in tissue differentiation as well.
Overall, the manuscript is well designed and carefully check all the experimental problems that could arise.
(1) I only have one concern in the analysis of RNAseq that I would like the authors to adress. The MA plot of Fig. 3A shows significant DEGs on the left side of the graph, identifying genes with low cpm. Typically, genes that are not highly expressed will be artefactually called DEGs in this portion of the graph. Normalization method often shrink this fraction of the transcriptome or even filter it out entirely to exclude low counts genes (see DESeq2 manual for example, even though the authors used edgeR). This is not specified in the methods.
Response: We thank the reviewer’s comments. According to the comments, we excluded genes with a normalized TMM value lower than 10 from the selected S-DEGs described in Table S3. The number of excluded genes were ten, all of which were male-specifically expressed genes (in other words, the normalized TMM value were zero). By this treatment, total number of S-DEGS were changed from 242 to 232. We performed re-analysis using the revised S-DEGs. As a result, correlation between the predicted expression levels of S-DEGs and the expression levels obtained by RNA-seq became stronger than the previous analysis. Note that this reanalysis did not change the number of outlier genes shown in Figure 3D. Please also see the lines 328-336, Table S3 and Figure 3E in the revised manuscript.
(2) Similarly, the alignment methods is not entirely described. I was concerned about the reference used to align. The authors mention (l. 175) the integration of two nucleotide sequences. Why ? What is the resulting reference file ? Can we have brief stats in a supplementary information on the number of genes in total, redundant or not, representing alternative splicing or not etc... ?
Response: We are sorry for insufficient explanations. All the publicly available reference sequence data in the silkworm, Bombyx mori, were established based on the data obtained from male genome. Thus, the reference sequence data did not contain female-specific sequence such as Fem gene derived from the W chromosome. To know the expression level of Fem was essential for this study, so we integrated the RefSeq generally used for RNA-seq analysis in the silkworm (https://ftp.ncbi.nlm.nih.gov/genomes/all/GCF/014/905/235/GCF_014905235.1_Bmori_2016v1.0/GCF_014905235.1_Bmori_2016v1.0_rna.fna.gz) with fasta data just containing only Fem RNA sequence (https://www.ncbi.nlm.nih.gov/nuccore/AB840787.1?report = fasta). AB840787 included in the URL corresponds to the NCBI accession number of the Fem gene. We added the above explanations to the revised manuscript. Please also see lines 175-181 in the revised manuscript.
Reviewer 3 Report
Comments and Suggestions for Authors
The manuscript "Transcriptomic evidence for cell-autonomous sex differentiation of the gynandromorphic fat body in the silkworm, Bombyx mori" provides a well-designed systematic analysis of sex differentiation process differences in silkworm. While the results are validated by quantitative RNA measurements, the study could be further improved by visualization of localization by immunostaining or in situ hybridization techniques if possible. Beyond this, there are only minor suggested changes, as follows:
1) Abstract, Line 23-24: Please briefly indicate/describe the ZZ and ZW chromosome system in the abstract to improve readability.
2) Introduction, Line 111: There is a typo on non-cell-autonomous
3) Figure 1 Legend, Line 244: Please reformat so the legend does not overrun into Figure 2.
4) Figure 3: Please revise Panel E left chart to remove the error line under RNA-seq.
5) The results of Figure 4-5 would be further improved and impactful if validated by some level of protein staining or even in situ RNA staining to visualize these stark localizations.
Comments on the Quality of English LanguageEnglish language is fine. A few minor typos were spotted, and the paper could thus use some editing to ensure all of them are corrected.
Author Response
We appreciate the many valuable comments. We have responded to each and every comment with the utmost sincerity. We hope that our response will satisfy the reviewer. Our alterations as a result of the reviewer’s comments are as follows.
The manuscript "Transcriptomic evidence for cell-autonomous sex differentiation of the gynandromorphic fat body in the silkworm, Bombyx mori" provides a well-designed systematic analysis of sex differentiation process differences in silkworm. While the results are validated by quantitative RNA measurements, the study could be further improved by visualization of localization by immunostaining or in situ hybridization techniques if possible. Beyond this, there are only minor suggested changes, as follows:
1) Abstract, Line 23-24: Please briefly indicate/describe the ZZ and ZW chromosome system in the abstract to improve readability.
Response: Sorry for insufficient explanations. According to the reviewer's comments, we added brief explanation about ZZ and ZW chromosome system in the abstract. Please also see lines 21-23 in the revised manuscript.
2) Introduction, Line 111: There is a typo on non-cell-autonomous
Response: We modified the typo according to the reviewer's comment. Please also see line 112 in the revised manuscript.
3) Figure 1 Legend, Line 244: Please reformat so the legend does not overrun into Figure 2.
Response: We thank the reviewer’s comment and reformat the position of Figure 2 so the legend of Figure 1 does not overrun into Figure 2. Please also see the page including Figure 2 in the revised manuscript.
4) Figure 3: Please revise Panel E left chart to remove the error line under RNA-seq.
Response: Sorry for the careless mistake. We removed the error line under RNA-seq. Please also see Figure 3E in the revised manuscript.
5) The results of Figure 4-5 would be further improved and impactful if validated by some level of protein staining or even in situ RNA staining to visualize these stark localizations.
Response: We agree with the reviewer's suggestion. Actually, we performed in situ hybridization to validate mosaic distribution of Fem and sp-1 expressing cells and to confirm co-expression of these two genes in the same cells in the gynandromorphic fat bodies. However, we struggled with high background presumably due to a very strong activity of endogenous alkaline phosphatase and were not able to recognize the localization of Fem and sp-1 mRNAs. For these reasons, if possible, please understand that this report will be submitted without visualization data such as ISH.